# Investigation of the Relationship between Morphology and Thermal Conductivity of Powder Metallurgically Prepared Aluminium Foams

**DOI:** 10.3390/ma14133623

**Published:** 2021-06-29

**Authors:** Arun Gopinathan, Jaroslav Jerz, Jaroslav Kováčik, Tomáš Dvorák

**Affiliations:** 1Institute of Materials & Machine Mechanics, Slovak Academy of Sciences, Dúbravská cesta 9, 845 13 Bratislava, Slovakia; ummsjerz@savba.sk (J.J.); ummsjk@savba.sk (J.K.); ummsdvor@savba.sk (T.D.); 2Faculty of Materials Science and Technology, Slovenská Technická Univerzita, Ulica Jána Bottu č. 25, 917 24 Trnava, Slovakia

**Keywords:** aluminium foam, closed-cell foam, thermal conductivity, X-ray tomography, modelling, powder metallurgy

## Abstract

Among different promising solutions, coupling closed-cell aluminium foam composite panels prepared by a powder metallurgical method with pore walls interconnected by microcracks, with low thermal conductivity phase change materials (PCMs), is one of the effective ways of increasing thermal conductivity for better performance of thermal storage systems in buildings. The internal structure of the foam formation, related to the porosity which decides the heat transfer rate, plays a significant role in the thermal energy storage performance. The dependence of the heat transfer characteristics on the internal foam structure is studied numerically in this work. The foamable precursor of 99.7% pure aluminium powder mixed with 0.15 wt.% of foaming agent, TiH_2_ powder, was prepared by compacting, and extruded to a volume of 20 × 40 × 5 mm. Two aluminium foam samples of 40 × 40 × 5 mm were examined with apparent densities of 0.7415 g/cm^3^ and 1.62375 g/cm^3^. The internal porous structure of the aluminium foam samples was modelled using X-ray tomography slices through image processing techniques for finite element analysis. The obtained numerical results for the heat transfer rate and effective thermal conductivity of the developed surrogate models revealed the influence of porosity, struts, and the presence of pore walls in determining the heat flow in the internal structure of the foam. Additionally, it was found that the pore size and its distribution determine the uniform heat flow rate in the entire foamed structure. The numerical data were then validated against the analytical predictions of thermal conductivity based on various correlations. It has been found that the simplified models of Bruggemann and Russell and the parallel–series model can predict the excellent effective thermal conductivity results of the foam throughout the porosity range. The optimal internal foam structure was studied to explore the possibilities of using aluminium foam for PCM-based thermal storage applications.

## 1. Introduction

Highly porous metallic structures with interconnected pores are a promising solution for enhancing the thermal conductivity of the phase change materials in TES applications [1]. Aluminium foam with a cellular matrix combines the benefits of the metal such as hardness, toughness, thermal and electrical conductivity with the functional properties of the foam such as stiffness, lightweight, adjustable density, and cellular structure [2]. The application of aluminium foam panels in the building sector for roofing and ceiling to bring better indoor comfort and thermal storage is important due to the better thermal transport properties of the foam. The foam structure is useful for various applications such as thermal transfer enhancement, insulation in buildings, etc. Such metal matrix composites with their high thermal conductivity to the coefficient of thermal expansion ratio bring the interest among many researchers to utilize in the building and various sectors where the need for better thermal performance arises [3].

In general, heat conduction in foams and cellular structures occurs through the solid skeleton and the fluid inside the 3D interconnected network of pore walls [4]. The equivalent thermal conductivity term, used for defining the performance of the foam in thermal applications, includes different heat transfer mechanisms. Effective thermal conductivity mainly refers to when a significant role is played by diffusion, and it controls the energy transport. In case of high temperature or in the application of foam having low thermal conductivity, radiative heat transfer plays a significant role in determining the equivalent thermal conductivity of the foam [4]. Moreover, the foam’s effective thermal conductivity is dependent on its constituent phase properties and the presence of porosity, and the formation of the internal structure of the foams. The parameters of the microstructure inside the foam and the properties of those porous structures are related qualitatively in most of the previous studies [5,6]. In all these studies, porosity is dealt with as the significant parameter influencing the porous structure’s thermal conductivity, and most of the empirical equations are based on porosity [6,7]. Apart from porosity, pore strut influence on thermal conductivity and the micropore effect are also studied [7,8].

Usually, the estimation of effective thermal conductivity of the metal foam can be done: (1) by conducting the experiments, (2) by numerical simulation using X-ray tomography images of the actual structure of the foam, (3) numerical simulation of idealized representations of foams and (4) by developing mathematical correlations that give the best fit with the experimental data [4]. Though the numerical heat transfer study of the open-cell foam structure has been studied many times [9,10,11,12], numerical study on thermal transport inside the closed-cell foam structure is minimal because of difficulty in modelling the complicated internal porous structure. Combining computer tomography image processing and 3D modelling software is an effective way of doing numerical modelling and finite element (FE) analysis [13,14,15,16,17,18]. It is highly efficient compared to the numerical simulation of uniform and realistic representations of 3D foam structures (e.g., cubic, tetrakaidecahedron, and Weaire–Phelan unit cells) as it doesn’t include the commercial foam irregularities and defects. Moreover, the non-destructive heat transfer simulation of the generated “surrogate models” from the X-ray slice images gives more information about the structural parameters of the internal structure of the foam and its influence on heat transfer enhancement, which is not possible in experimental and mathematical findings.

The present work aims to characterize the produced closed-cell aluminium foam with interconnected pore walls and ruptures through X-ray tomography and develop a numerical 3D surrogate model of the same, which helps predict the thermal performance of the internal structure of the foam in the FE analysis method. The aluminium foam produced through the powder metallurgy (PM) technique has many internal spherical pores and pores walls; the discretization helped represent that the internal pore structures are not viable and lead to significant computation times. To overcome this problem, computer tomography of the samples is performed. Commercially available software such as 3D slicer, ImageJ, Meshmixer, and Ansys has been used in this study to create surrogate models and for the clear visualization of the internal porosity and broken pore walls responsible for the heat conduction inside the foam. The resulting models are then subjected to a steady-state heat transfer study across the thickness of the foam. The obtained heat transfer rate results are used to find the effective thermal conductivity of the surrogate models with different porosity ranges and are then validated analytically with the available predictions of various literature.

## 2. Methodology

### 2.1. Preparation of Samples

The powder metallurgical process prepares aluminium foam samples for testing according to the procedure described by Simančík et al. [19]. The samples are prepared from an aluminium powder of 99.7% purity with a particle size of <63 µm. A more accurate particle size distribution is determined using the Retsch Analysette 22 dry laser diffraction method (Institute of Materials and Machine Mechanics—Slovak Academy of Sciences, Bratislava, Slovakia). The analysis showed the median particle diameter of the powder to be d_50_ = 43 µm (e.g., 50% particles is smaller than d_50_), d_10_ = 21 µm and d_90_ = 79 µm. It is reported by the powder supplier KERAMETAL, Ltd., Slovakia, that the aluminium powder used in this study is composed of 99.7 wt.% Al, 0.11 wt.% Fe and 0.06 wt.% Si. Based on that, the solid thermal conductivity of the Al (99.7 wt.% purity) chosen for the work, is 225.3 W/(m·K) [20]. The use of this fine-grained powder to prepare a foamable precursor has been shown to contribute significantly to preventing the foam structure from collapsing during foaming when the aluminium foam is melted. A foamable precursor is prepared to contain only 0.15 wt.% of foaming agent, TiH_2_ powder, to achieve higher density of the foam more easily [21]. 

The final mixture of the powder is subjected to cold isostatic compaction under 200 MPa of pressure. Billets of 30 mm dimension (approx.) were produced. The billets were hot extruded to a profile with a rectangular cross-section (5 × 20 mm^2^). The extrusion temperature was set to 450 °C at an extrusion ratio of 28:1. The sample specimens were foamed in a steel mould in an electric resistance furnace in the form of small square plates (40 × 40 × 5 mm) using the foamable precursor of dimensions 20 × 40 × 5 mm. The furnace temperature for sample preparation was maintained at 720 °C. Ten samples were foamed with good reproducibility in the density range of 0.7 to 1.8 g/cm^3^. 

The foamable rectangular profile and the aluminium foam sample produced are shown in Figure 1. The porosity is calculated by using the following equations [22].
(1)$$Porosity=1-\frac{\rho^{*}}{\rho_{s}}$$
(2)$$Porosity=\frac{V_{p}}{V_{T}}$$
where $\rho^{*}$, $\rho_{s}$ & $\rho_{r}$ are the apparent, bulk, and relative densities. $V_{p}$ & $V_{T}$ are the pore volume and total volume of solid and pore. Equation (1) is used to find the porosity when the apparent density of the sample is known. For 3D models, Equation (2) is utilised to find the porosity by comparing the models with the solid model of the exact dimensions. Two samples A1 and A2 were chosen for the numerical investigation from all foamed samples, and their porosity was calculated based on Equation (1). The corresponding structural properties can be found in Table 1. 

As most of the metal foam structure volume is occupied by pores and the volume proportion of the walls is less, the thermal conductivity of the metal foams is generally lower than that of the solid aluminium metal. The thermal conductivity of the aluminium foam is greatly influenced by the density of the foam [23]. The foamable parts are usually covered by an aluminium oxide layer of a few microns thickness, giving them a metallic appearance and a better surface area for heat conduction.

### 2.2. X-ray Tomography of Aluminium Foam

Observation of the structure was done using an Phoenix/X-ray Nonatom 180. CXZ device (Manufacturer—G & E, Institute of Materials and Machine Mechanics—Slovak Academy of Sciences, Bratislava, Slovakia). This approach is a suitable method for studying the pore wall architecture and distribution of metal and pores. The slice images made quantitatively helps to calculate the tortuosity of the different phases, density distribution, pores, and cell size distribution [24]. The detailed depictions of the aluminium foam for the 3D model were obtained from the CT images of the aluminium foams. The number of slice images made on the XY plane, the YZ plane, and the XZ plane and the obtained resolution at 96 × 96 dpi is tabulated in Table 2. 

The CT slice images made at the midsection of the samples are shown in Figure 2. A set of CT slice images can be used for the numerical investigation of the thermal conductivity of actual foams.

## 3. Modelling and Analysis

### 3.1. Creation of Model

Metal foam has a highly complex pore structure, and it is made with a combination of randomness and irregularity [15]. The foam properties are based on the pore structure, and the complicated structure is challenging to interpret the morphology of the foam [15]. A 3D graphic model is developed from the obtained CT slice images by detecting the edges of the images. To perform it, 3D slicer software is utilized for this study. The 3D slicer is used to visualize the whole 3D model of the foam sample. Making a realistic model suitable for finite element analysis gives more information about the heat transfer process occurring through the internal structure than a 3D model made from mathematical structures such as cubes, tetrakaidekahedrons and dodecahedrons. Moreover, it would be beneficial to visualize and understand the morphology, and the thermal transport that occurs in the various structure formed. 

The procedure involves creating an X-ray image stack from the series of CT images by importing them into the 3D slicer software. The volume rendering option controls the coordinates to visualize the live mask 3D model of the stacked slice images. The CT-MIP display option is chosen, and it gives the metallic texture, which is shown in Figure 3.

On comparing with the real-time model, the accuracy of the geometrical conformity is less. The development of the 3D model results from automatic reconstruction by keeping the default smoothening factor of 0.5 and the scale factor as 1. Using the threshold option in the segment editor module, the live mask 3D model can convert to an “.stl” file, which can be imported and used for conducting numerical studies with Ansys. Since the created models have many elements and complicated structures, it is difficult to export as a “.stl” file or “.obj” file. Moreover, it is time-consuming to import and simulate the whole 3D model of the aluminium foam with Ansys software. For this reason, it is proposed to split the actual foam sample into smaller sections, and to perform numerical simulations on each model [18].

### 3.2. Recreation of the Model

The 3D model of the sectional aluminium foam is recreated with the stacks of cropped sectional slices of the CT tomographic in the 3D slicer software, and the model created is crucial for the accurate description of the heat transfer of the closed-cell foam. The 3D model is then exported as a “.stl” file. The scale factor is kept constant to 1 in the 3D slicer software.

The reduced modelling procedure is explained with the A1 and A2 sample. As the structure inside the foam is quite complicated to simulate and the computation time is longer, the whole foam sample is divided into 16 sections of equal dimensions 9.8 × 9.8 × 5 mm (approx.). The crop volume function of the 3D slicer software is utilized for this purpose. The modelling accuracy depends on the distance between slicing planes when the X-ray tomography images were made, which affects the minimum level of the detectable pore size and the complexity of pore wall structures [25]. As the steady-state heat transfer process is studied in one-dimension, the thickness of the foam is kept constant for all the models at 5 mm. This approach helps find the function of foam struts, pores, and breakage of pore walls formed inside the closed-cell foam structure in the heat conduction process at a particular area.

Moreover, it should be needed to visualize the influence of non-uniform material deposition and density distribution all over the aluminium foam in determining the effective thermal conductivity of the foam, ultimately a deciding factor for determining the heat storage with PCM at a particular section of the foam. Furthermore, a solid model of the same dimensions was created for the heat transfer study to differentiate the difference in heat transfer between the foam models with porosity and without porosity. The cropped sections made are depicted in the midsection slice images in Figure 4.

The computational domain of the sole solid phase was regenerated by the foam reconstruction procedure in Ansys workbench software. The steady-state analysis module of the Ansys workbench was utilised, and the models were then first imported to the SpaceClaim geometry module. The models created in the 3D slicer software are of the dimensions based on the slice images imported. Since the model size is different from the actual dimensions, the scaling option was used in the SpaceClaim module of Ansys to keep the thickness as 5 mm. This was achieved by using the measure tool, and the thickness of the imported foam model was measured. The scale factor was set by dividing the actual thickness of the foam (5 mm) by the thickness of the foam measured after importing it into the SpaceClaim. The value of the thickness of the foam measured is the same as that of the number of cropped slice images.
(3)$$\mathrm{Scale}\;\mathrm{factor}=\frac{L}{L_{S}}$$
where L is the actual thickness of the sample and $L_{S}$ is the thickness of the foam measured after importing into the SpaceClaim. In the model section, several mesh options have been tried on the surface of the foam for high-quality refinement [10]. Due to the complexity of the geometry, creating a highly structured mesh is complicated, leading to a dense mesh at the porous structure. To overcome this, the body sizing option and the fluent preference settings give the tetrahedral cells, which provided a better solution for remeshing the mesh model for reconstruction and reproducing all the details of the foam’s geometry. The steps followed in creating the model are explained with the flowchart, and a detailed description is given of the creation of model number A2 × 11 shown in Figure 5.

The typical models transformed have different properties, e.g., mass, volume, presence of pores and number of elements. As the value of porosity also differs, the models are imported to Meshmixer to find the volume of the created models. On comparing the volume of the generated meshed model with the volume of the solid 3D model of the exact dimensions, the porosity of the sectional model was found. The total porosity generally includes the material pores, cavities, pores and pore walls [26].

### 3.3. Boundary Conditions and Simulations

To determine the temperature distribution of the aluminium foam, a one-dimensional steady-state analysis is performed along the direction of the foam model of thickness 5 mm with appropriate boundary conditions. The Ansys workbench steady-state module is used for the simulation process—pre-processing, solving, and post-processing. The initial temperature was set as default 22 °C. As the application of the aluminium foam is considered for roofing which is directly exposed to sunlight in the Bratislava region, Slovak republic, it was assumed that the plate was facing the South directly to analyse the maximum point, and the maximum amount of energy calculated in the irradiance calculator was 5.36 kWh/m^2^/day recorded in June [27]. From the energy value, the temperature at the top surface of the plate was calculated as 63.85 °C. The temperature at the bottom surface of the plate was kept at room temperature (say 22 °C). It was assumed that the heat transfer occurs only through the conduction of the solid pore walls.

Steady-state thermal analysis, in general, is used to calculate the effect of steady thermal load on the system. The analysis is used to determine the temperature, thermal gradient, heat flow rate, and the amount of heat flow which does not vary with time. Steady-state analysis was adopted in this work to study the temperature and amount of heat flow that occurs when the aluminium foam is subjected to a steady thermal load. The reaction probe tool was placed at the top and bottom surface of the model to find the amount of heat (Q) transferred. The temperature contour of the foam model A2 × 11 simulated in the Ansys workbench is presented in Figure 6 below.

## 4. Analytical Approach and Validation

The effective thermal conductivity of the foam is a term that is used to refer to the case when diffusion plays a role [19]. It is an intrinsic solid-phase conductivity that is not as same as that of the bulk material thermal conductivity. The decrease in bulk thermal conductivity has been found experimentally when the foam is produced [2,23]. The higher effective thermal conductivity of the metal foam results from the lower porosity of the foam with a higher thermal conductivity metal matrix, having lower conduction resistance. It has been noted that the effective thermal conductivity of the metal foam is mainly dependent on porosity. However, the independence of the pore size and the interconnected pore walls present inside the structure in depicting the effective thermal conductivity is not clear [19].

The numerical simulation performed in this work is validated with the previously described analytical approach of various heat transfer studies on the foam to ensure the current finite element simulations [13]. In general, thermal conductivity ‘λ’ and thermal diffusivity ‘a’ are the two material properties that characterize heat conduction in solids. According to Fourier’s law of heat conductivity, the heat flux, q, induced by a temperature gradient ∇T, is:(4)q=− λ·∇T

The units of λ are J/(m·s·K) or W/(m·K) [28]. The general equation for finding the effective thermal conductivity of the foam, $\lambda^{*},$ is given by [13],
(5)$$\lambda^{*}=\frac{Q\cdot L}{\nabla T\cdot A}$$
where $\lambda^{*}$ is the effective thermal conductivity (W/(m·K)), Q is the heat transfer flow rate (W) [29], A is the surface area perpendicular to the direction of heat flow (m^2^), and L is the thickness of the sample (m). The heat conduction surface area on model, A, is given as,
A = A_T_ – A_P_(6)
where A_T_ is the total cross-sectional area and A_P_ is the area occupied by the pores [14].

A wide variety of models proposed for closed-cell foams are available, and the most acceptable model is given by Ashby et al. [5]. From [5], the thermal conductivity of the foam $\lambda^{*}$, could be thought of as having four contributions: conduction through the solid (Aluminium composite), $\lambda_{s}^{*}$; conduction through the gas trapped inside the pores, $\lambda_{g}^{*}$; convection within the pores, $\lambda_{c}^{*}$; and the radiation from the pore walls across the voids of the pores, $\lambda_{r}^{*}$. Hence the thermal conductivity is given by the sum of all four [5].
(7)$$\lambda^{*}=\lambda_{s}^{*}+\lambda_{g}^{*}+\lambda_{c}^{*}+\lambda_{r}^{*}$$

The conductivity of the solid is given by the following expressions [5].
(8)$$\lambda_{s}^{*}=\eta \cdot \lambda_{s}\cdot \left(\frac{\rho^{*}}{\rho_{s}}\right)$$
where $\lambda_{s}$ is the conductivity of the fully dense solid aluminium composites, $\frac{\rho^{*}}{\rho_{s}}$ is the volume fraction of the solid (*V_f_*) or the relative density, and η is the efficiency factor, which refers to the tortuous (fibrous) shape of the pore walls. For closed-cell foam, the value of η is 2/3 for the tortuous shape of the pore walls. It firmly fits the present simulation results.

To find the influence of the convection in the pore walls, the pore length is calculated according to the procedure mentioned by Ashby et al. [6]. The following simplified equation calculates the pore length, *l*.
(9)$$l=\sqrt[{3}]{\frac{1000\;\cdot \mu^{2}\cdot \;T}{g\;\cdot \Delta T_{c}\cdot \rho_{g}^{2}}}$$
where g is the acceleration due to gravity (9.81 m/s^2)^, T is the temperature of the foam due to the applied heat, Δ*T_c_* is the temperature difference between the cell or pore walls, and $\rho_{g}$ and μ are the density and dynamic viscosity of the gas, respectively. From the numerical simulation, the minimum temperature difference experienced between the cell walls in the foam (Δ*T_c_)* is 9.336 K, and the inside gas is assumed to be air ($\rho_{g\left(air\right)}$ = 1 kg/m^3^ and μ = 2 × 10^−5^ N·s/m^2^). From Equation (9), the minimum pore length calculated is 11.37 mm by setting the Grashoff number to 1000. Since the pore length of the foam of both the samples is less than 11.37 mm, the convective heat transfer is negligible compared to solid-phase conduction, and it is not considered [6]. 

The thermal conductivity through radiation is calculated by the following simplified equation [6].
(10)$$\lambda_{r}^{*}=4\cdot \;\beta_{1}\;\cdot \;\sigma \;\cdot {\overset{-}{T}}^{3}\;\cdot \;L\;\cdot {\;e}^{\left[-\left(\frac{\rho^{*}}{\rho_{s}}\right){\mathrm{LK}}_{s}^{*}\right]}$$
where, $\beta_{1}$ is the constant less than unity, σ is Stefan’s constant (5.67 × 10^−8^ W/(m^2^·K^4^)), $\overset{-}{T}$ is the mean temperature ((T_1_−T_0_)/2), L is the thickness of the foam and $K_{s}^{*}$ is the extinction coefficient of solid aluminium. For foam sample A1, the thermal conductivity through radiation was calculated as 0.03538 W/(m·K). Since the thermal conductivity through radiation decreases with an increase in relative density, it was assumed to be negligible.

The simplified model for heat conduction through solids is given by Solorzano et al. [2]. As the simplified equation is based on solid-phase conduction ($\lambda_{s}^{*}>>\lambda_{g}^{*})$, it is considered and admitted to the present work. The five models suggested are considered in this study based on the porosity value [2]. The assumption was made $\left(\lambda^{*}=\lambda_{s}^{*}\right)$ in which only the conduction through the solid phase would be considered. The various theoretical models considered for this study are tabulated in Table 3.

## 5. Results

### 5.1. Pore Size and Morphology

Most of the pore space formed inside the closed-cell aluminium foam is spherical and closed fully or partially in which the breakage of the walls and interconnection between the pores occurs [1]. ImageJ image processing software is usually utilized to analyse the size of the pores, pore spacing, and count. By analysing the X-ray tomography images of the foam, no pores of definite shape were found, and pore wall ruptures were visible in both samples. The prediction of pore wall distance with the slice image itself is complex. When the pore wall distance is measured in the ImageJ software, and the same pore is measured in the SpaceClaim module of the Ansys workbench with the 3D model of the pores at the midsection, a specific length difference can be observed due to the 3D structure (see Figure 7). The measure tool is used to calculate the distance between two vertices of the pore wall.

By comparing the measured values, this approach seems to be promising to calculate the depth of the pore and the pore wall distance present inside the foam at any point with both slice images and a 3D model. The same approach is utilised by using the threshold option in the ImageJ software to find the pore space area distribution at the midsection slice image, and the results for pore distribution of A1 and A2 samples are given in Figure 8. The graph is plotted by keeping the pore area (mm^2^) along the X-axis and the number of pores along the Y-axis. The pore distribution is arranged into the 10 classes with the pore space area shown in the graph below (Figure 8). The pore space distribution at the midsection is affected by considering the interconnected pores as one pore [1].

The development of the 3D model helps to understand the inner pore wall structure formation in a non-destructive way. For example, Meshmixer software is utilised to study and visualize the formation of pores and pore wall ruptures in 3D view.

After importing the model into the Meshmixer software, it is then aligned at the base point, and it distinguishes the solid and the inner pore structure, which is shown in Figure 9. The cropped 3D model of the A1 and A2 samples and the formation of pores inside the structure are shown in Figure 10 below.

From Figure 10, the A1 foam with higher porosity has a large pore space area which is significantly less in the case of the lower porosity foam, A2. At the same time, the presence of micropores is higher in the case of the lower porosity foam A2 compared to the higher porosity foam A1. This approach could help visualize the presence of pores, how significant the presence of the pores is, the presence of ruptures, etc. Though it gives a more realistic view of the inhomogeneous foam structure produced in real-time, modelling of the defects that occurred in the pore walls is poor. It can be controlled by selecting the appropriate threshold value of the slice images for generating the 3D model.

### 5.2. Numerical Study

The thermal conductivity of the solid phase foam is generally lower compared to the thermal conductivity of pure bulk aluminium due to the porosity present inside the internal structure and the negative influence of oxide impurities in the strut material [26]. A thin layer of the aluminium oxide layer formed on the surface of the aluminium foam, and the thermal conductivity of the Al_2_O_3_ (24–39 W/(m·K)) decreases with an increase in temperature above room temperature [26,30]. Therefore, the aluminium oxide layer formed on the surface is assumed for the present work. As mentioned, the numerical study conducted is mainly carried out in one direction, and the thickness of all the foam models created is kept constant (5 mm). The quantity of the heat passing through the base area of the samples, Q obtained through numerical simulation, is used to calculate the effective thermal conductivity. The numerical and analytical results’ effective thermal conductivities are compared, and the closest one is considered to be the best fit in this study.

The typical models, specific surface area and volume are obtained by importing the models in the Meshmixer. The porosity of the surrogate models is calculated using Equation (2). The amount of heat transferred from the one-dimensional steady-state analysis is found using Ansys simulations, and effective thermal conductivity is calculated using Equation (5). The results are tabulated in Table 4 below. A heat transfer study of the solid model of the same dimension as the foam model is used as the reference. 

The surface area, A, which is perpendicular to the heat flow, is the summation of the average area of the faces selected at one side where the heat is applied since the STL file has the number of faces. The area of the single meshed face of the “.stl” model ranges from 0.13–0.135 mm^2^. This approach neglects the pore area at the surface of the model. The face area of the samples is maintained very low at the reconstruction step to keep the micropores and ruptures. Since the surface area value perpendicular to the heat flow changes based on the number of pores at the surface of the foam, the calculated effective thermal conductivity values differ for some foamed models having the same porosity.

From the results in Table 4, the effective thermal conductivity values are normalised by the thermal conductivity of solid Al, 225.3 W/(m·K) [2]. The sectional models developed from both the samples are of different porosity levels in the range of 8% to 70%. Since the analytical models are restricted to the solid thermal conductivity, some of the simplified analytical models give the same expressions, and the thermal conductivity values are the same. The overall results numerical and analytical results based on the equations mentioned in Table 3 are shown in Figure 11.

The empirical formulae mentioned in the literature for predicting thermal conductivity show some merit. The series and the parallel model mentioned in Table 3 are not added to the graph as the calculated results are non-realistic when the solid thermal conductivity is considered. So that they are considered not suitable for the thermal conductivity predictions of the foam model developed. Simplified effective medium theory-based thermal conductivity predictions give very low thermal conductivity when the porosity increases above 40% and give negative values when the porosity reaches above 66%. So that the effective medium theory is not suitable for the thermal conductivity predictions for foams above 40%, and it is also excluded.

On the contrary, the series–parallel and Misnar predictions and the Ashby model fit the models with porosity above 50%. Below 50%, those model predictions are very low. By comparing the other available models based on porosity such as Doherty, Maxwell and Eucken, the Bruggeman model with 1/3 fit for spherical pores [31], the simplified model of Russell and the parallel–series model give a close representation of effective thermal conductivity values compared to the obtained numerical data in the porosity range. 

## 6. Discussion

### 6.1. Reconstruction of Actual Foam and Defects

The thermal conductivity of the foam increased dramatically when the porosity reduced. There are several reasons for the errors of the different models, especially for the higher porosity models, which lack the support of the present systematic research. The possible explanations for the error factors are related to the accuracy of the reconstruction process of the .stl file in which the minimum level of the detectable pore size is affected due to the distance between the slicing planes when the X-ray tomography image is made [25]. In general, higher resolution of X-ray tomography enables better modelling of foam irregularities and their influence on thermal conductivity.

Secondly, the apparent thermal conductivity value depends not only on the presence of porosity level but also on the shape of the cell, the density distribution, the connectivity of the pore walls, and other geometric imperfections such as pore wall ruptures with microcracks present, pore walls misalignments, fractured walls, size variations of the pores, etc., [2,32]. It is visible in Figure 12 below. Moreover, it is evident from the heat transfer values of models A1 × 32, and Al × 42 mentioned in Table 4. Though both models have the same porosity level, the amount of heat transfer is higher for A1 × 42 s than for A1 × 32 due to various factors such as density distribution, pore wall thickness, and strut thickness.

### 6.2. Influence of Struts and Pore Wall Thickness

The total heat flux has been checked at each foam model to find the influence of struts and thin ligament distribution on the amount of heat transfer apart from porosity. Every surrogate model created, which is different from the idealized representation of the model, has different ligament thicknesses. Unlike the solid model showing a linear heat flux rate, the porous 3D models created show non-linear heat flux rates inside the structure, which increases at the thin ligaments, at the merged surfaces of struts and nodes, and the thin cell walls. As mentioned in [10], the merged surfaces of struts and nodes constitute considerable solid elements predominant in the structure of closed-cell foams. The heat conduction is limited to the small strut cross-sectional area [10], which is higher in the case of higher porosity foams where bigger pores are more likely to be present with thin struts. This is the reason for the reduction in the average heat flux rate and lower foam thermal conductivity. At the same time, the maximum total heat flux rate has been found at the struts and the merged strut and node areas since the cross-sectional area of them is lower, as shown in Figure 13a,b.

Since thin ligament formation is low in the lower porosity foams, the amount of heat flow occurring is linear for the most part of the inside structure compared with the higher porosity foams (Figure 14b).

Furthermore, the models with the same porosity level but different pore size distribution were investigated to find the influence of the pore size and distribution of pores in determining the heat flow rate. This can be seen in the surrogate models A1 × 11 and A2 × 22 (with porosity of 41%) in which the A1×11 has wider pore distribution, whereas the A2 × 22 has narrower pore distribution. The average heat flux obtained for both the models was 1.52 W/mm^2^.

However, the heat flux is not the same for both the models. The A1 × 11 model experience highly non-uniform heat transfer rate at the cell walls (Figure 14a) in comparison with the A2 × 22 model (Figure 14b). The results show the importance of pore size and its distribution of the foam in PCM based heat storage applications where uniform heating is required.

### 6.3. Comparison with Analytical Models

The numerical results obtained depend mainly on the porosity of the structure which makes it meaningful to validate them with the available analytical models having the correlations based on the porosity and morphological parameters. Variations in the structural parameters of the pores and their distribution inside the actual foam structure limits the use of analytical models that have correlations based only on morphological parameters for predicting the effective thermal conductivity of closed-cell foam [4]. Further, the analytical models based on the porosity have different correlations with the assumption of uniform pores distribution inside the structure, which is inaccurate. The sizes of the pores are not uniform and they of various shapes, which has a substantial impact on the thermal conductivity of the closed-cell foam. This is the opposite of the previous predictions done with open-cell foam of uniform pore size distribution [32].

There are no single empirical correlations that can define the thermal conductivity of the aluminium foam produced by the PM route in the whole porosity range [2]. It can be again concluded that the Bruggemann, Russell and parallel–series model predictions coincide best with the observed numerical data throughout the porosity range. Moreover, those models are of interest in this study for predicting the actual foam thermal conductivity, since the lower porosity foams are the major consideration for the further future study on PCM based heat storage applications. 

Overall, the approach used for investigating the actual foam is very helpful to find out the optimum porosity level, inner porous structure for better heat transfer and PCM-based heat storage applications.

## 7. Conclusions

This primary research is focused on the influence of the internal porous structure formation of aluminium foam in enhancing the heat transfer rate under steady-state conditions. A methodology has been developed to perform FEM simulations of heat transfer in the complex 3D structure of the foam materials with structural inhomogeneities and defects. The novelty in the developed methodology is the introduction of the image processing technique in developing a sectional 3D model of different porosities by transforming a cropped image stack of the foam sample. Using this approach, the surrogate models of porosity in the range of 8% to 70% were developed. In the first phase of the study, the developed methodology helped to find the morphology of the pores and the distribution of the pores present inside the foam. The measured pore wall distance at a particular pore in both slice image and the 3D model had very good agreement. This proves the efficiency of the developed methodology in finding the structural parameters of the foam and the pores in a non-destructive way. 

In the second phase, the heat transfer simulation under steady-state conditions was conducted numerically. The heat flux developed at the struts and pore walls was studied in addition to the porosity. The results show the importance of thickness changes for uniform heat transfer.

Furthermore, the obtained effective thermal conductivity values are compared with the analytical models. The simplified classical models, such as series, parallel, and effective medium theory, did not show a good fit when solid thermal conductivity alone was considered. Simplified series–parallel and Misnar predictions and Ashby’s model showed good agreement only for porosities above 50%. The models such as the simplified model of Russell, Parallel–Series and Bruggemann’s model have shown the best agreement with numerically predicted thermal conductivity values throughout the porosity range. Apart from porosity, the effect of defects and the density distribution of metal within foam structure was considered which was found to be one of the main sources for the differences in obtained results. Furthermore, the methodology used shows the potential to optimize foam structure for the development of PCM-based TES systems.

## Figures and Tables

**Figure 1 materials-14-03623-f001:**
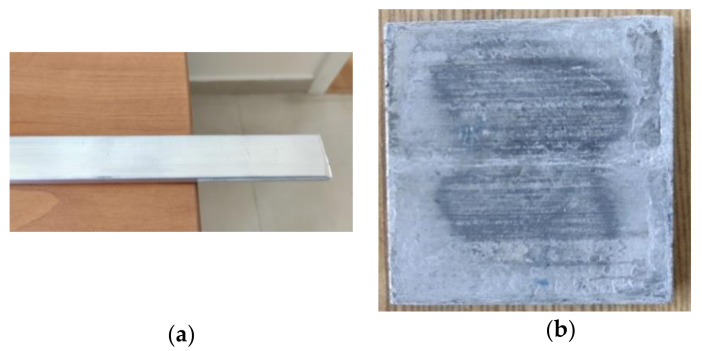
(**a**) Foamable rectangular profile of cross-section (5 × 20 mm^2^) and (**b**) Aluminium foam sample prepared (40 × 40 × 5 mm).

**Figure 2 materials-14-03623-f002:**
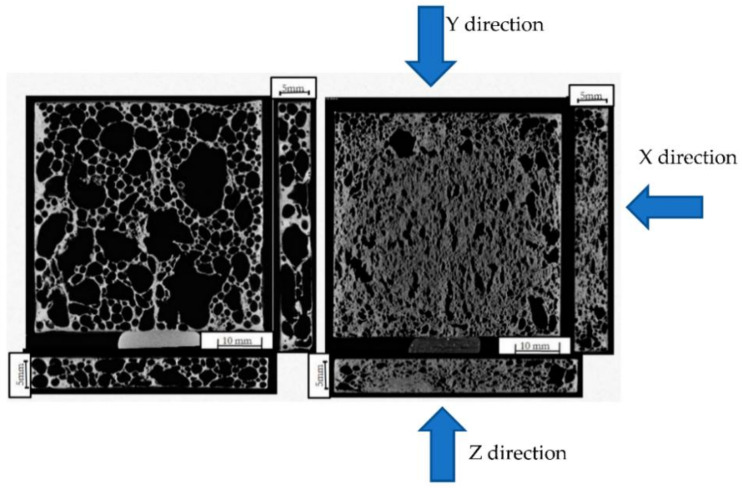
Typical CT slice image of the A1 and A2 aluminium foam sample at the midsection (X direction—right side view, Y direction—front view & Z direction—bottom view).

**Figure 3 materials-14-03623-f003:**
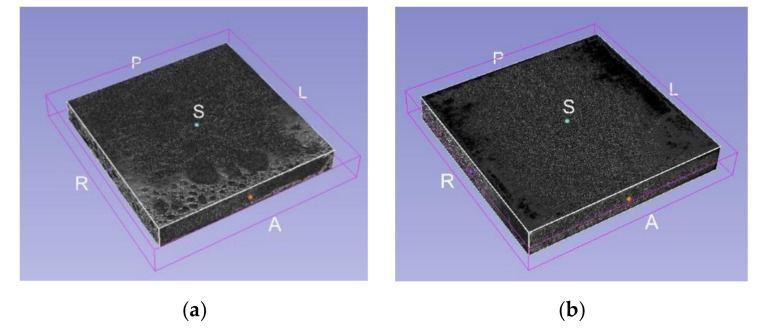
3D model of the stacked CT images of Aluminium Foam sample A1 (**a**) & A2 (**b**) (40 × 40 × 5 mm).

**Figure 4 materials-14-03623-f004:**
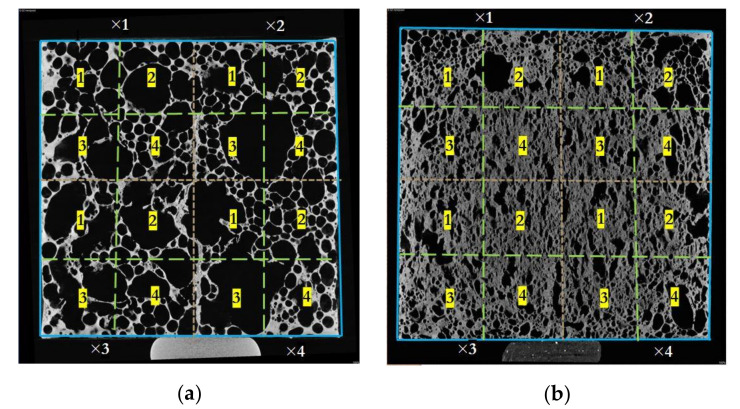
16 split sections are represented with the dotted line of the sample A1 (**a**) and A2 (**b**). The X-ray slice images are cropped according to these split sections, and the 3D model is constructed.

**Figure 5 materials-14-03623-f005:**
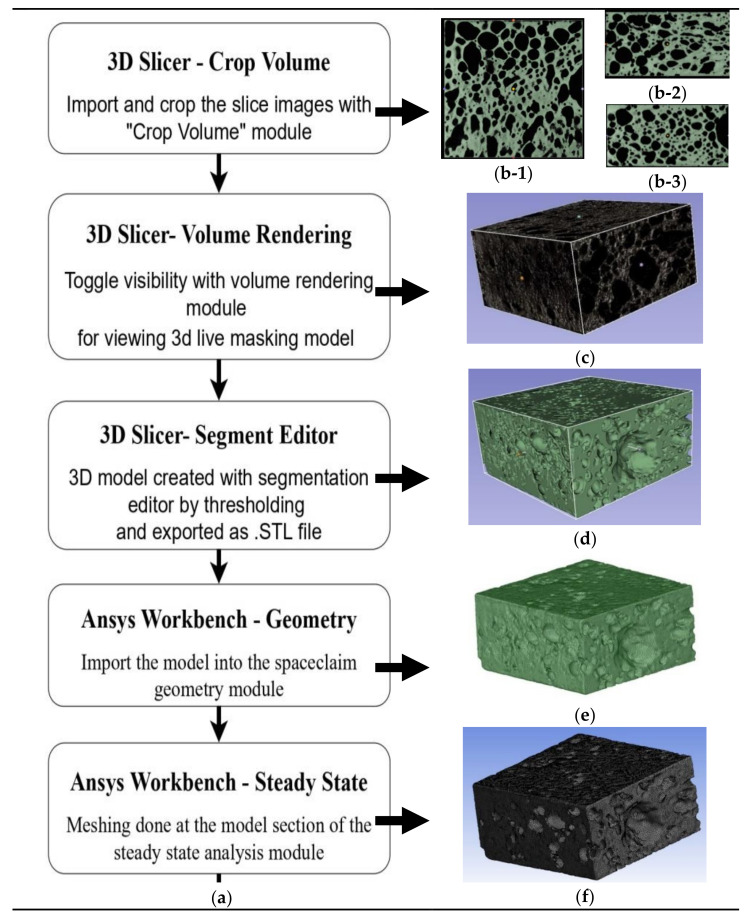
Steps involved in the creation of the model. (**a**) flow chart, (**b-1**–**b-3**) axial, sagittal, and coronal plane views of cropped X-ray stacking slice images (3D slicer), (**c**) live masking 3D model (3D slicer), (**d**) generated 3D model as “.stl” file(3D slicer), (**e**) reconstructed model (Ansys workbench) and (**f**) meshed model (Ansys workbench).

**Figure 6 materials-14-03623-f006:**
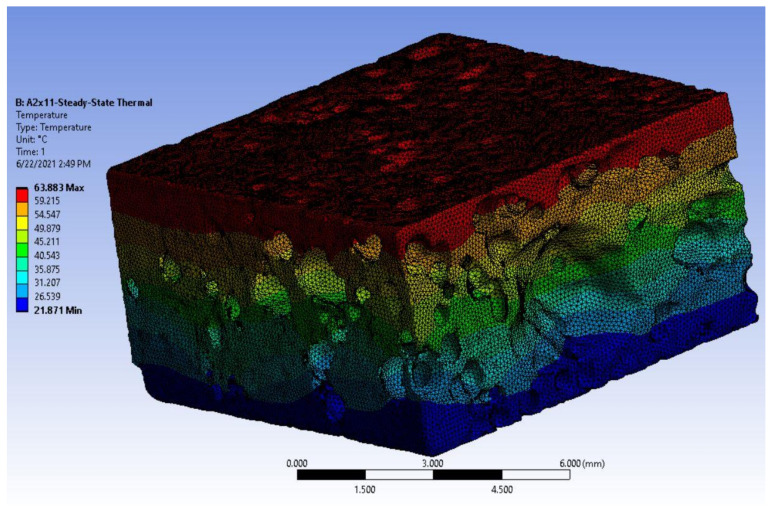
Steady-state one-dimensional conductive heat transfer of the foam model A2 × 11.

**Figure 7 materials-14-03623-f007:**
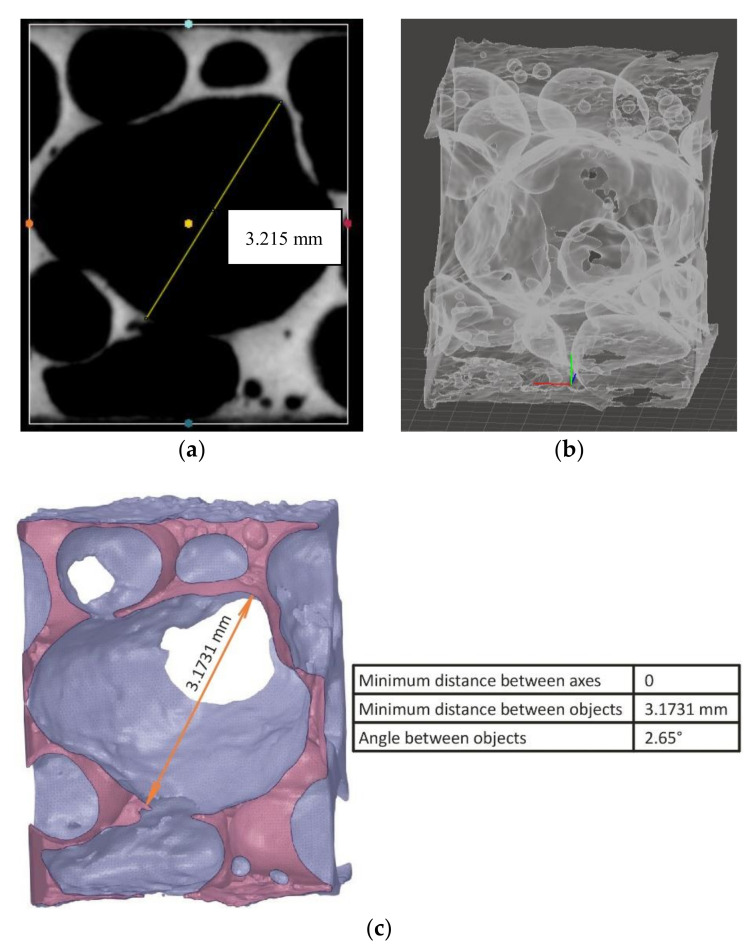
(**a**) Pore wall distance of 3.215 mm measured at ImageJ software of the sagittal image view of the midsection slice image, (**b**) pore distribution, and (**c**) pore wall distance of 3.1731 mm measured at the midsection of the 3D model in SpaceClaim module of Ansys workbench.

**Figure 8 materials-14-03623-f008:**
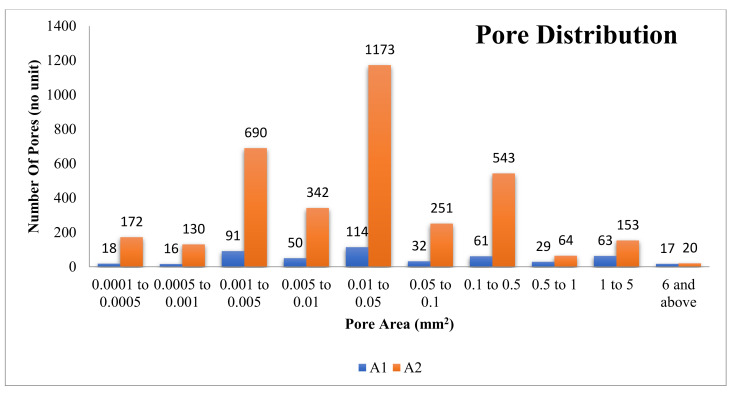
Distribution of pore area (mm^2^) and the number of pores inside the structure of the A1 and A2 models.

**Figure 9 materials-14-03623-f009:**
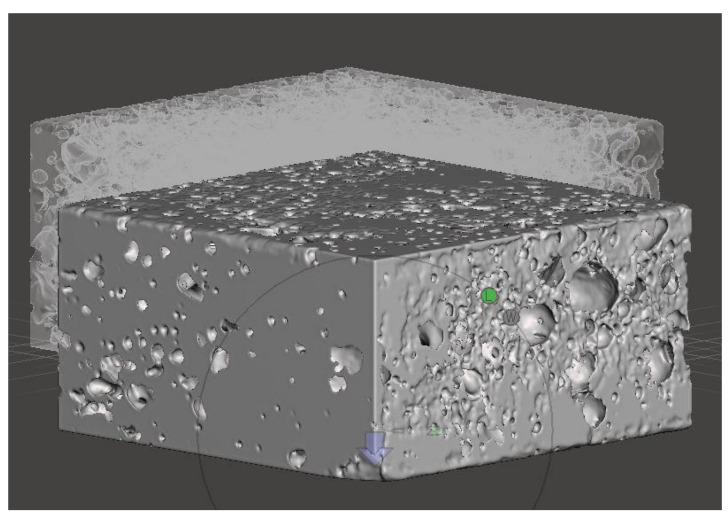
Meshmixer align tool distinguishing the solid and pore structure presence of the “.stl” file of the A1 × 11 3D model.

**Figure 10 materials-14-03623-f010:**
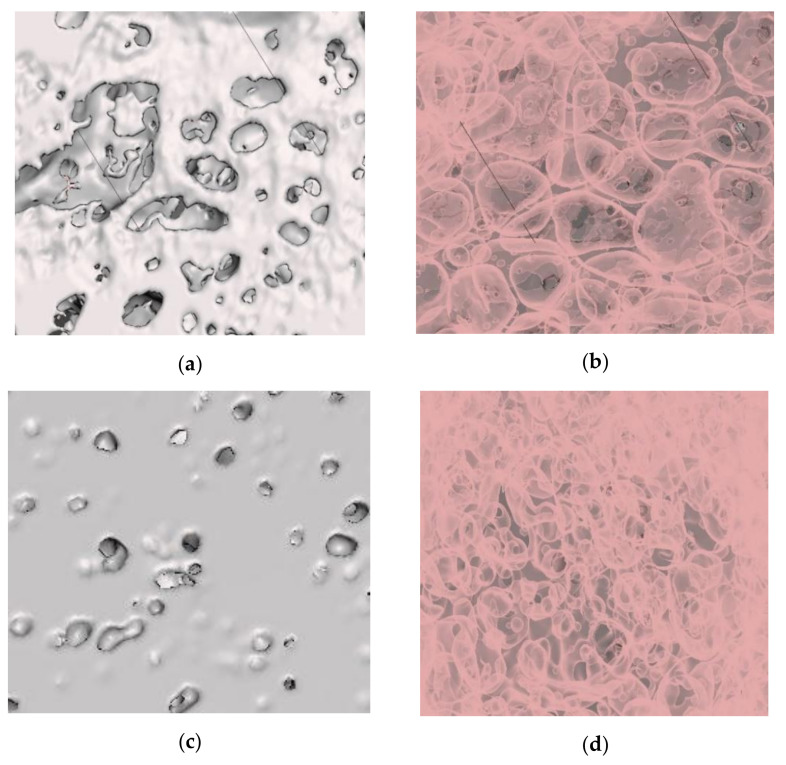
Cropped 3D model. Top view of foam, and the distribution of pores inside the 3D model (identical view), with 72.5% porosity (**a**,**b**) and with 39.9% porosity (**c**,**d**).

**Figure 11 materials-14-03623-f011:**
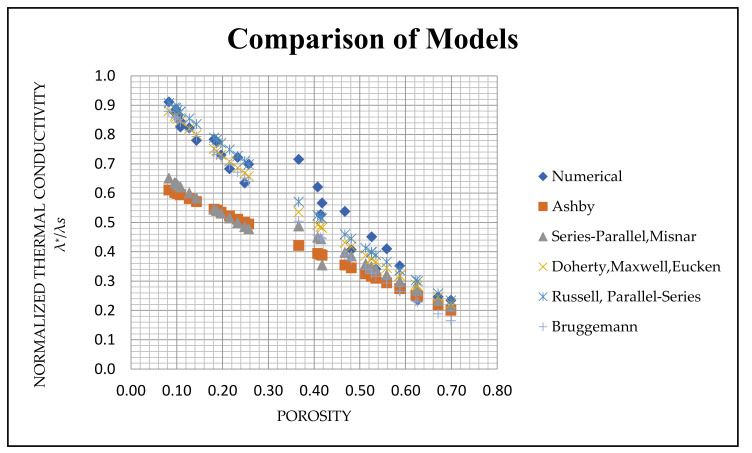
Normalised thermal conductivity predictions of the theoretical models from Table 3, as a function of the foam porosity in comparison with the numerical results ($\lambda_{s}=225.3\;W/\left(m\cdot K\right))$.

**Figure 12 materials-14-03623-f012:**
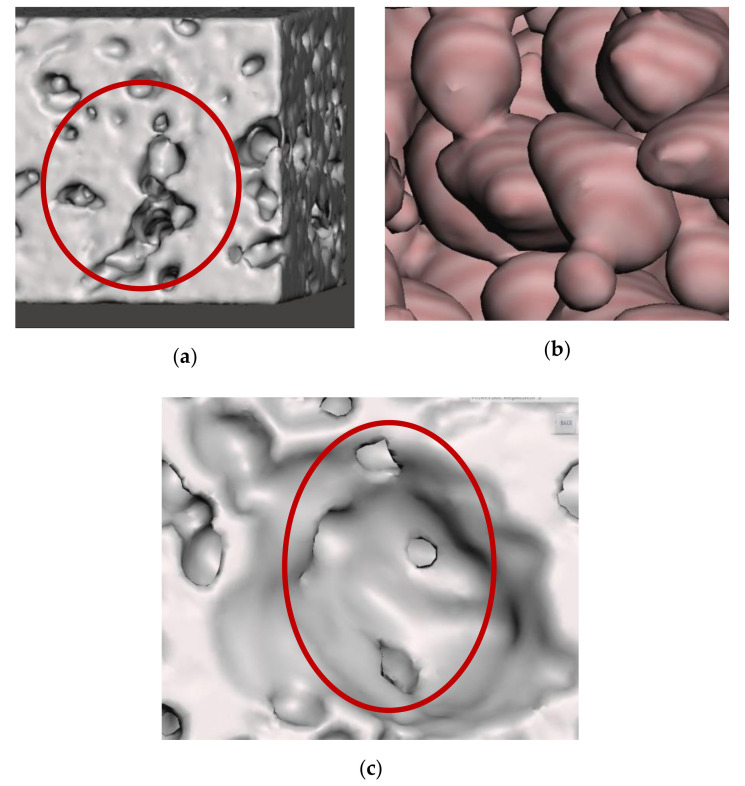
Structure of the foam (**a**) continuation of pore walls, (**b**) pores of different size joined inside the foam, and (**c**) pore wall defects.

**Figure 13 materials-14-03623-f013:**
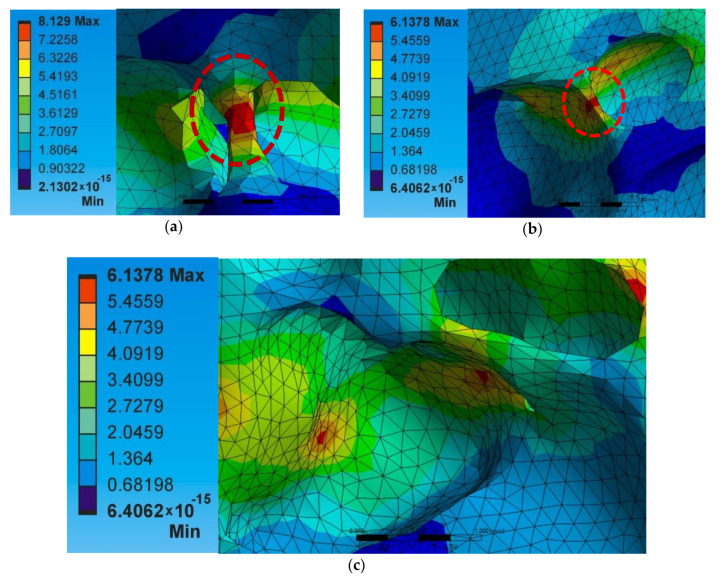
Maximum heat flux (W/mm^2^) at (**a**) thin ligaments (A1 × 32) and (**b**) struts (A1 × 42), and (**c**) the heat flux distribution at the cell walls (A1 × 42)—52% foam porosity.

**Figure 14 materials-14-03623-f014:**
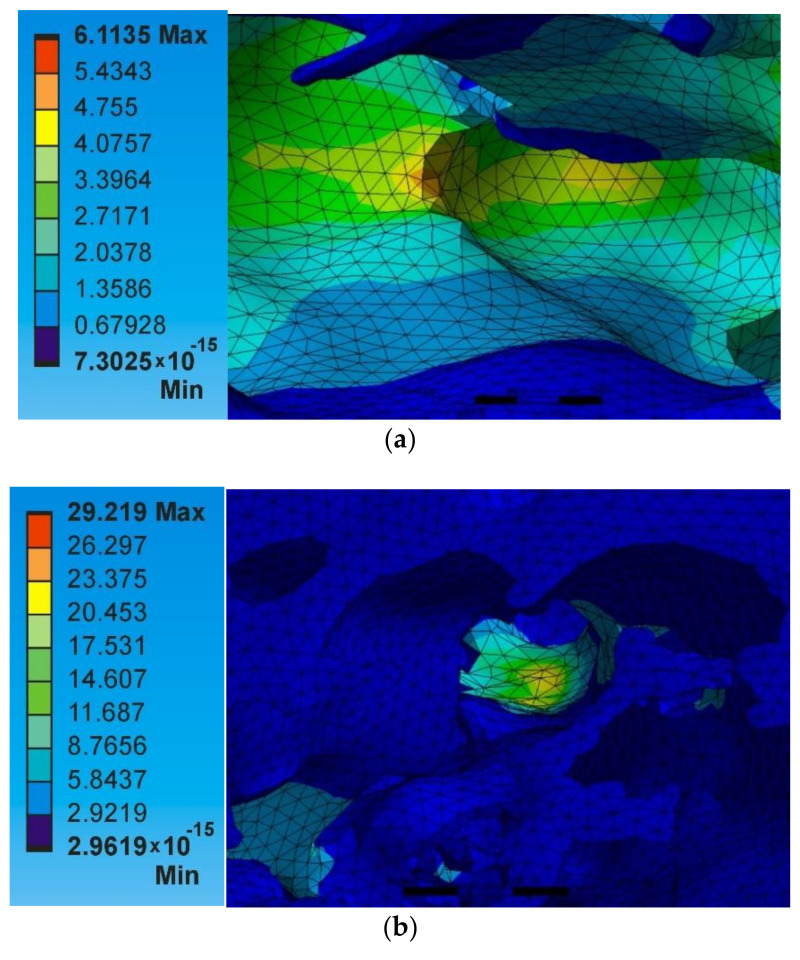
Heat flux (W/mm^2^) distribution at the cell walls of the models with same porosity value (**a**) A1 × 11 and (**b**) A2 × 22—41% foam porosity.

**Table 1 materials-14-03623-t001:** Structural properties of the aluminium foam samples.

Sample (Al 99.7%)	$\mathrm{Apparent}\;\mathrm{Density},\;\rho^{*}\;(g/{\mathrm{cm}}^{3})$	$\mathrm{Relative}\;\mathrm{Density},\;\rho_{r}$	Porosity (%)
A1	0.7413	0.2745	72.5
A2	1.62375	0.6014	39.9

**Table 2 materials-14-03623-t002:** The number of CT slice images made along planes and their resolutions (approx).

CT Slice Images	Plane		
	XY	YZ	XZ
Sample A1-images	1828	275	1846
Resolution (mm)	207 × 575	551 × 575	551 × 207
Sample A2-images	1846	257	1830
Resolution (mm)	169 × 553	526 × 553	526 × 169

**Table 3 materials-14-03623-t003:** Theoretical models under consideration [2,7].

Literature Model	Original Equation
Series	λs*=1(Vf/λs)
Parallel	$\lambda_{s}^{*}=\lambda_{s}*V_{f}$
Ashby	$\lambda_{s}^{*}=\eta \lambda_{s}\left(\frac{\rho^{*}}{\rho_{s}}\right)$
Series–Parallel	λs*=λs1−Vf23+λsVf23λg+(λs−λg)Vf13
Parallel–Series	λs*=λsλs−(λs−λg)Vf23λs−(λs−λg)(Vf23−Vf)
Maxwell	λs*=λg2Vf+1+λsλg21−Vfλsλg1−Vf+2+Vf
Doherty	$\lambda_{s}^{*}=\frac{\lambda_{g}\lambda_{s}\left(2V_{f}+1\right)+\left(2\lambda_{s}^{2}\left(1-V_{f}\right)\right)}{\lambda_{g}\left(1-V_{f}\right)+\lambda_{s}\left(2+V_{f}\right)}$
Eucken	λs*=λs1+2Vf[1−λsλg/2λsλg+1]1−Vf[1−λsλg/2λsλg+1]
Russell	λs*=λgVf23+λsλg1−Vf231−Vf23+Vf+λgλsVf23−Vf
Bruggemann	1−Vf=λg−λs*λg−λsλsλs*1/3
Misnar	λs*=λs1+1−λsλg1−Vf131−λsλg
Effective Medium Theory	$(1-V_{f})\frac{\lambda_{s}-\lambda_{s}^{*}}{\lambda_{s}+2\lambda_{s}^{*}}+V_{f}\frac{\lambda_{g}-\lambda_{s}^{*}}{\lambda_{g}+2\lambda_{s}^{*}}=0$

**Table 4 materials-14-03623-t004:** Nomenclature, structural parameters, numerical heat transfer and effective thermal conductivity values of the models.

Sectional Model	Total Surface Area, A_T_ (mm^2^)	Volume, V_s_ (mm^3^)	Porosity, (−)	Amount of Heat Transferred, Q(W)	$\mathrm{Effective}\;\mathrm{Thermal}\;\mathrm{Conductivity},\;\lambda^{*}\;(W/(m\cdot K))$
A1 × 11	746.279	281.363	0.4141	82.386	118.7901
A1 × 12	829.780	223.401	0.5348	55.770	76.9624
A1 × 13	911.695	179.458	0.6263	38.101	53.1681
A1 × 14	983.216	144.769	0.6985	33.079	52.9591
A1 × 21	735.981	284.441	0.4077	82.632	140.0198
A1 × 22	605.721	304.306	0.3663	102.06	161.3040
A1 × 23	895.662	158.188	0.6706	34.423	55.6175
A1 × 24	707.148	234.158	0.5124	50.337	76.5937
A1 × 31	806.345	198.646	0.5867	50.624	79.3599
A1 × 32	859.840	228.164	0.5249	54.934	74.7945
A1 × 33	863.694	255.886	0.4671	78.903	121.2035
A1 × 34	797.629	249.219	0.4810	69.839	91.7156
A1 × 41	816.204	198.064	0.5875	47.265	64.5991
A1 × 42	777.008	227.724	0.5258	60.863	101.7067
A1 × 43	765.947	181.521	0.6220	39.695	57.0227
A1 × 44	762.215	212.093	0.5583	58.670	92.4256
A2 × 11	981.644	356.867	0.2568	119.09	157.4956
A2 × 12	711.010	393.237	0.1811	136.08	176.6529
A2 × 13	785.134	411.559	0.1429	138.40	175.8803
A2 × 14	683.321	434.460	0.0953	147.79	197.1705
A2 × 21	833.184	390.388	0.1870	134.57	174.8835
A2 × 22	1176.810	279.846	0.4172	82.143	127.7012
A2 × 23	713.388	428.457	0.1078	139.33	186.1727
A2 × 24	916.761	377.033	0.2148	114.07	154.0352
A2 × 31	752.608	419.181	0.1271	142.89	185.4161
A2 × 32	653.556	440.510	0.0827	157.01	205.3113
A2 × 33	828.518	385.874	0.1964	124.79	164.6494
A2 × 34	697.840	428.582	0.1075	148.94	190.5596
A2 × 41	693.305	431.242	0.1020	145.28	194.7236
A2 × 42	985.347	361.126	0.2480	107.52	143.0932
A2 × 43	656.713	433.204	0.0979	149.36	199.7562
A2 × 44	869.328	368.362	0.2329	119.83	162.7146
Solid model	388.08	480.2	0	181.11	225.3

## Data Availability

No new data were created or analyzed in this study. Data sharing is not applicable to this article.

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
