# Peer review of "Investigation of the Relationship between Morphology and Thermal Conductivity of Powder Metallurgically Prepared Aluminium Foams"

_materials, 2021, doi:10.3390/ma14133623_

Round 1

Reviewer 1 Report

In the present manuscript, the relationship between morphology and thermal conductivity of an aluminium foam was investigated by FEA based on Ansys workbench cooperated with 3D slicer, ImageJ, Meshmixer. Unfortunately, the manuscript is not well-written. I do not recommend the publication of this work in Materials.

Here are some detailed comments:

  1. This manuscript was prepared just like a tutorial on how to use software rather than a research article.
  2. The authors did not carefully check this paper result in numerous errors in the manuscript, such as disappearance of spacing, inconformity of the style of reference, and other mistakes (incorrect symbol in the explanation of equation.9).
  3. Discussion section is not included.

The conclusion is senseless. In fact, using computer simulation, it is more valuable to investigate the properties (size, shape, curvature, distribution, knots, et. al) of pores/walls how to influence the thermal conductivity of aluminium foam. 

Author Response

Respected Sir/Madam,

Thank you for your valuable time and we are happy to get your comments. The comments have helped us to improve the manuscript.

The manuscript is rewritten by adding the discussion chapter and the following areas are concentrated.

  1. the heat flux at the struts and pore walls are investigated
  2. heat flux variation based on the size, shape, and distribution of the pores at the same porosity level is added
  3. the manuscript is corrected and an initial grammar check is done.

In addition, the Abstract, introduction, experimental procedures of producing the samples, and the conclusion is rewritten

All the changes made are marked in the manuscript and the revised manuscript is submitted for your review.

Thank you once.

With regards,
Arun Gopinathan

Reviewer 2 Report

This manuscript proposes a study on the relationship between morphology and thermal properties of the metal foams obtained by using a foaming agent. Interesting results and a numerical model are proposed. The paper needs to be revised before publication.

  • Introduction is poor and must be rewritten;
  • Do not use multi reference [4,5,6], [7-10], [3,11,12,13,14-16], [5,6,17,18]. Use 1 or 2 citations giving more details on the specific results;
  • Why the authors have chosen a precursor with a contain of 0.15wt.% of foaming agent – TiH2 powder? Some reference could help to justify the choice. The porosity distribution depends on the foaming agent content.
  • A better description of the experimental setup to obtain the foams is appropriate
  • How many specimens have produced the authors? Is there a repeatability in the foam morphology?

Author Response

Respected Sir/Madam,

Thank you for your valuable time and we are happy to get your comments. The comments have helped us to improve the manuscript.

According to the comments,

  1. Introduction have rewritten
  2. Multiple references are removed and appropriate citations are added
  3. The usage of 0.15wt.% of foaming agent – TiH2 is to achieve the high-density foam structure and the appropriate reference is added for justification
  4. The experimental procedure for foam production is explained 
  5. Totally 10 samples are produced in the density range of 0.7 g/cm3 to 1.8 g/cm3 with reproducibility. This information is furnished

In addition, the discussion chapter and the following areas are concentrated.

  1. the heat flux at the struts and pore walls are investigated
  2. heat flux variation based on the size, shape, and distribution of the pores at the same porosity level is added
  3. the manuscript template is corrected and an initial grammar check is done.

All the changes made are marked in the manuscript and the revised manuscript is submitted for your review.

Thank you once. Looking forward to hearing from you.

With regards,
Arun Gopinathan

Reviewer 3 Report

The paper presents the determination of thermal conductivity of  2 types of aluminum foams.  The 3D reconstruction of foam structures is a demanding task for engineers. The Authors have shown an useable method using computer tomography. The heat transfer finite element model of the problem is established in a pleasant way. The strength of the paper is the comparison of the analytical calculation with the numerical results.

With the structure constructed by geometric reconstruction, the actual geometry can be examined. On the one hand, this is accurate, and on the other hand, it is difficult to generalize the result obtained due to the reproducibility problems of aluminum foams. Therefore researchers are on to develop the so-called surrogate models. 

However, the paper is interesting and up-to-date, it needs some corrections, listed below:

  • there are spellings in the text (even in the abstract, e.g. TiH2powder; 5mmhave; g/cm3and);
  • Equation 1, 2 and 3 contain words that need to be eliminated and the Authors should use symbols;
  • in Figure 6.a the size is not visible;
  • in Table 4 more columns without name;
  • surrogate modell has to be mentioned in the text.

Author Response

Respected Sir/Madam,

Thank you for your valuable time and we are happy to get your comments. The comments have helped us to improve the manuscript.

According to the comments,

  1. The spellings and spaces are checked in the whole manuscript
  2. Equation 1, 2, and 3 are rewritten
  3. The size in figure 6 is made to be visible
  4. Table 4 is corrected 
  5. Importantly, the term surrogate models are used 

In addition, the discussion chapter is added and some changes are made

All the changes made are marked in the manuscript and the revised manuscript is submitted for your review.

Thank you once. Looking forward to hearing from you.

With regards,
Arun Gopinathan

Round 2

Reviewer 1 Report

In the present manuscript, the influence of porosity, struts and pore walls presence in deciding the heat flow at the internal structure of the aluminum foam prepared via powder metallurgically was well investigated by FEA based on Ansys workbench cooperated with 3D slicer, ImageJ, Meshmixer. Thus, the reviewer suggests this manuscript to be published in Materials after the authors can address the following issues:

  • To easily understand the microstructure of the aluminum foam, the directions (X, Y, and Z) should be defined.
  • It is more preciseness and more artistic if replace the ‘X’ with ‘’ in equation (10) to follow the previous formulas.
  • Are Figure 7a and 7c in the same position? They are similar in morphology, however, some small pores seem to have been abnegated in the 3D model of workbench. Is it caused by the resolution of x-ray tomography, data import of 3D slicer software, recreation of 3D model or meshing in workbench? What fallowed is the accuracy of Figure 8?
  • It is unfriendly to readers that the color bars or necessary notes on colors of Figure 13 and 14 are forgotten.

5)    There are at least four types of references in this manuscript, and some of them are listed at the end of this question. All references should be replaced in a format not only including whether abbreviation of the publication titles, because it is necessary for an eligible article.

  1. Duschlbauer, D.; Böhm, H. J.; Pettermann, H. E. Numerical Simulation of Thermal Conductivity of MMCs: Effect of Thermal Interface Resistance, Materials Science and Technology 2003, vol. 19, pp. 1107–1114.
  2. Gibson, L. J.; Ashby, M. F.; Zhang, J.; Triantafillou, T. C. Failure surfaces for cellular materials under multiaxial loads – I. Modelling. Int. J. Mech. Sci.1989, 31(9), pp. 635–663.
  3. Nammi, S. K.; Myler, P.; Edwards, G. Finite element analysis of closed-cell aluminium foam under quasi-static loading. Materials & Design 2010, Volume 31, Issue 2, pp. 712–722.
  4. Sutygina, A.; Betke, U.; Scheffler, M. Open-Cell Aluminum Foams by the Sponge Replication Technique. Materials (Basel), 2019, 12(23): 3840.

Author Response

Respected Sir/Madam,

Thank you for your comments and for your valuable time. The manuscript is revised according to your comments.

  • The direction x,y,z is added with the image for understanding
  • The X symbol is replaced on how it is mentioned in previous equations
  • Image 7c is changed. The problem is due to the recreation of the model when the split body option is used for making the cross-section of the solid body. It is rectified by making the cross-section on the faceted body where the exact morphology is made to be visible. 
  • The color bars of the heat flux is added with figures 13 and 14
  • References are corrected and the full abbreviation of the journal is provided

Please kindly review it and provide your valuable comments.
Thank you once.

With regards,

Arun

Reviewer 2 Report

The authors have revised the paper according the reviewer suggestions.

The paper can be accepted in present form.

Author Response

Thank you